# The Pseudo-Protic Ionic Liquids TOAH^+^Cl^−^ and TODAH^+^Cl^−^ as Carriers for Facilitated Transport of In(III) from HCl Solutions

**DOI:** 10.3390/membranes13010019

**Published:** 2022-12-23

**Authors:** Francisco José Alguacil, Félix Antonio López

**Affiliations:** Centro Nacional de Investigaciones Metalurgicas (CENIM-CSIC), Avda. Gregorio del Amo 8, 28040 Madrid, Spain

**Keywords:** facilitated transport, TOAH^+^Cl^−^, TODAH^+^Cl^−^, pseudo-protic ionic liquids, indium, hydrochloric acid

## Abstract

A study of indium(III) transport across an immobilized liquid membrane using the pseudo-protic ionic liquids TOAH^+^Cl^−^ and TODAH^+^Cl^−^ as carriers has been carried out using batch experiments. Metal transport is investigated as a function of different variables: hydrodynamic conditions in the feed (375–1500 min^−1^) and receiving (500–750 min^−1^) phases, HCl (0.5–7 M) and indium (0.01–0.2 g/L) concentrations in the feed phase and carrier (1.25–40% *v*/*v*) concentration in the membrane phase. Indium is conveniently recovered in the receiving phase, using a 0.1 M HCl solution. Models are reported describing the transport mechanism, which consists of a diffusion process through the feed aqueous diffusion layer, fast interfacial chemical reaction, and diffusion of the respective indium-pseudo-protic ionic liquid through the membrane. The equations describing the rate of transport are derived by correlating the membrane permeability coefficient to diffusional and equilibrium parameters as well as the chemical composition of the respective indium-pseudo-protic ionic liquid system, i.e., the carrier concentration in the membrane phase. The models allow us to estimate diffusional parameters associated with each of the systems; in addition, the minimum thickness of the feed boundary layer is calculated as 3.3 × 10^−3^ cm and 4.3 × 10^−3^ cm for the In-TOAH^+^Cl^−^ and In-TODAH^+^Cl^−^ systems, respectively.

## 1. Introduction

At the time at which this manuscript is being written (November 2022), there is in industrialized countries a general concern about the regular supply of metals to fulfil various industrial necessities. These metals are needed due to their use in smart technologies: communications, computers, batteries, electronics, etc. The doubtful supply of strategic metals, such as lithium, niobium, tantalum, rare earths, indium, etc., means that academia, governments, private research institutions, private companies and entrepreneurs try to check for new sources in which these strategic metals can be found. This is because the concept of urban mining and the recycle of its wastes has been becoming more popular for a long time.

Particularly, indium appeared in several urban devices in the form of ITO (indium tin oxide). This oxides mixture, when deposited as a thin film on glass or clear plastic, is used in several display technologies, such as LCD, OLED, plasma, and electroluminescent and electrochromatic displays, as well as in several touch screen technologies.

In the treatment of these wastes, hydrometallurgy emerged as a real solution to solve the problem of the treatment of these residues, and the recovery of the valuable metals encountered in them. Generally speaking, and after the pre-treatment of the waste by various operations, including (i) dismantling, (ii) shedding and (iii) comminution, a leaching step totally or partially dissolves the metallic component of the waste. From the leachate, very often containing more than one metal species, different separation technologies (liquid–liquid extraction, ion exchange, adsorption) are used to separate (and sometimes also concentrate) one metal from another, before to the final recovery step of the valuable metal or metallic compound.

Liquid membrane processes have been of increasing interest; however, at present, not a single process has known industrial application. The main liquid membrane operations include supported and unsupported liquid membranes. While the latter is currently of minor interest, the former consists of a thin microporous and hydrophobic polymer support in which the carrier phase (the organic phase containing the extractant) is immobilized. This support separates the feed and receiving or stripping solutions. First batch investigations in the form of flat-sheet configuration can be scaled up to a continuous form in hollow fibre modules. From an engineering and practical point of views, supported liquid membranes are of particular interest due to their stability (if correctly operated) and simplicity, because they combine in a single operation the extraction and stripping stages [1].

In the particular case of indium, recent data about the use of the above separation technologies includes:

(i)Liquid–liquid extraction. D2EHPA (di-2-ethylhexyl phosphoric acid) is used to extract indium(III) [2]; however, difficulties found in the stripping stage mean that the use of certain additives is recommended to facilitate this operation. Among the additives used, 2-ethylhexanol presented the best characteristics for its practical use. Spent copper indium gallium selenide (CIGS) materials have been found to be a source for the recovery of indium and galium [3]. After some previous steps, the leaching solution was extracted with P204 (di-2-ethylhexyl phosphoric acid) extractant for the sequential removal of indium and gallium; both metals are recovered from the respective organic phase using HCl as a strippant. Differing in their number and size of coordination sites, calix[4]arene, alkenyltrimethylol, and trihydroxytriphenylmethane frameworks are prepared to extract indium(III) [4]. The calix[4]arene and trihydroxytriphenylmethane derivatives demonstrated extraction of In^3+^ with an unexpected stoichiometry of 1:2. In the treatment of spent liquid crystal displays (LCDs) of monitors, a process which includes microwave pyrolysis, leaching and solvent extraction is investigated [5]; D2EHPA is again used as an extraction agent, whereas the stripping stage is carried out with 6 M HCl as a strippant solution. In another investigation using also real solutions from the treatment of spent LCDs [6], Cyanex 923 (industrial phosphine oxides mixture) is used to extract iron(III) and indium(III) from the leachate. Separation of both elements is performed, varying the V_aq_/V_org_ phases relationship; firstly, iron is extracted using a high relationship, and after, indium(III) is extracted at a low ratio, i.e., 1. A real sulphate solution obtained by leaching of dross is subjected to a solvent extraction operation to separate germanium and indium [7]. After the extraction of germanium with a trioctyl amine and tributyl phosphate mixture, indium is extracted with D2EHPA and stripped in a hydrochloric acid medium. 15-crown-5 (15C5), benzo-15-crown-5 (B15C5), and thia-benzo-15-crown-5 (TB15C5) extractants are investigated for selective indium recovery from end-of-life liquid crystal displays (LCDs)’ leaching solution [8]. In this case, the theoretical and experimental extraction order is 15C5 > B15C5 > TB15C5. Various hydrometallurgical processes involving pressure acid leaching and solvent extraction are investigated for the recovery of indium from zinc slag, which is produced in the imperial smelting process [9]. Pressure leaching with sulfuric acid is the preferred method for recovering indium from the slag, and again, indium is extracted with D2EHPA and stripped in HCl medium.(ii)Adsorption. Metal-organic frameworks (MOFs) of UiO-66 are used in the adsorption of indium at acidic pH values [10]. Results show that the adsorbent structure is stable even at these acidic pH values, and that together with an adsorption process, a cation exchange mechanism is involved in indium uptake onto the adsorbent.(iii)Ion exchange with resins. After the leaching of indium from waste liquid crystal displays with HCl solutions, the metal is recovered from the corresponding leachate with an anion exchange resin [11]. This resin is a strongly basic (with quaternary ammonium functional groups on polystyrene-DVB matrix) gel-type resin (Varion AD) in chloride form. Spent LCD screens are leached with sulfuric acid in order to recover indium [12]. The leachate is then treated with ion exchangers to separate indium from other metallic impurities. Best results are obtained with Lewatit VP OC 1026, a D2EHPA impregnated resin, and the corresponding metal elution is carried out with HCl solutions. Lewatit TP 208 was impregnated with different extractants to enhance its properties in the removal of indium from diluted solutions [13]. The impregnation of the resin with D2EHPA and Cyanex 272 (organic alkyl phosphinic acid derivative) not only improves the selective indium recovery; additionally, the resin capacity was approximately doubled. The resin impregnated with Cyanex 272 presented the best results with respect to indium uptake onto the resin and elution.(iv)Membranes. A bipolar membrane electrodialysis (BPED), using ethylenediaminetetraacetic acid (EDTA) as a chelating agent, is used to separate indium(III) from aluminium(III) [14]. In the above medium, indium forms the anionic complex InY^−^ (Y represents the chelating agent), whereas aluminium is present as Al^3+^.

With respect to ionic liquids, much has been written about this compounds and uses. It is important to remember that the term of ionic liquids includes those chemicals that are composed only of ions (anion and cation), which are liquids at temperatures below 100 °C. Due to their properties of negligible vapour pressure, non-molecular solvents, high thermal stability, adjustable viscosity and miscibility with organic solvents and water [15], ionic liquids are considered *green solvents*; however, some authors questioned some of these properties, i.e., stability [16]. A special case of these compounds was the pseudo-protic ionic liquids (PPILS) formed from tertiary amines [17,18].

The aim of the present work is to estimate parameters to optimize the performance of a flat-sheet supported liquid membrane impregnated with TOAH^+^Cl^−^ and TODAH^+^Cl^−^ PPILs, in the active transport of indium(III) from HCl solutions. Different experimental variables are evaluated in the transport of indium(III) by these carriers, and from the experimental data, diffusional parameters are calculated for both systems, In(III)-HCl-TOAH^+^Cl^−^ and In(III)-HCl-TODAH^+^Cl^−^.

## 2. Materials and Methods

### 2.1. Materials

The precursors for the generation of the PPILs was the tertiary amines Hostarex A324 and Hostarex A327 (Frankfurt del Main, Germany) was formed by tri-octyl amine (TOA), whereas Hostarex A327 was composed of a 50% mixture of tri-octyl and tri-decyl amines (TODA). The reagents were diluted in Solvesso 100 (aromatic diluent, ExxonMobil Chemical Europe, Machelen, Belgium) in order to achieve an adequate range of amine concentrations and therefore enough of the ionic liquid for the indium transport experiments. The corresponding PPILs TOAH^+^Cl^−^ and TODAH^+^Cl^−^ were formed by contacting the organic phase (amine and diluent) with adequate HCl solutions. After equilibrium, the ionic liquids were formed in the organic phase, according to the next equilibrium [19,20]:(1)$$R_{3}N_{org}+H_{aq}^{+}+Cl_{aq}^{-}\Leftrightarrow R_{3}NH^{+}Cl_{org}^{-}$$
where the subscripts org and aq represent the respective organic and aqueous phases. For both amines, the percentage of conversion to the pseudo-protic ionic liquid exceeded 99%. The use of an organic diluent, such as Solvesso 100, was recommended due to the fact that it allowed (i) an adequate the range of carrier concentrations in the investigation, and (ii) a decrease in the viscosity of the ionic liquid and thus, a reduction of the resistance to transport due to the organic phase.

1 g/L indium(III) stock solution was prepared by dissolving an indium salt (Fisher Scientific S.L., Madrid, Spain) in distilled water, and working solutions were prepared from dilution of the above. All the chemicals used in the experimentation, except the amines and Solvesso 100, were of A.R. grade. The solid support used in the present work was Millipore Durapore GVHP4700 (polyvinylidene fluoride) of 75% porosity, 1.67 tortuosity and 12.5 × 10^−3^ cm thickness.

### 2.2. Methods

Transport experiments were performed in a two compartment cell, consisting of a feed phase (200 cm^3^) separated from the receiving phase (200 cm^3^) by the membrane support. The effective membrane area for the transport experiments was of 11.3 cm^2^. Both feed and receiving phases were mechanically stirred by four-blade glass impellers (2.5 cm diameter) at 20 °C, to avoid concentration polarization conditions at the support interfaces and in the bulk of both phases [21].

The supported liquid membrane was prepared by impregnation of the solid support with the corresponding organic phase, by immersion for 24 h. It was then left to drip for 20 s before it was placed in the membrane cell. Previous tests demonstrated that extended immersion times did not influence indium transport.

The percentage of indium transported from the feed phase to the membrane phase was determined by monitoring the metal concentration in the feed phase at elapsed times using atomic absorption spectrometry (Perkin Elmer 1100B spectrophotometer, Waltham, MA, USA), and using the following relationship:(2)$$\%T=\frac{{\left[In\right]}_{f,0}-{\left[In\right]}_{f,t}}{{\left[In\right]}_{f,0}}\times 100$$
where [In]_f,0_ and [In]_f,t_ were the indium concentrations in the feed phase at time zero and at elapsed time, respectively. The percentage of indium recovered, from the membrane phase to the receiving phase, was calculated using the following relationship:(3)$$\%R=\frac{{\left[In\right]}_{r,t}}{{\left[In\right]}_{f,0}-{\left\|In\right\|}_{f,t}}\times 100$$
where [In]_r,t_ represented the indium concentration in the receiving phase at an elapsed time.

In the cases in which the permeation coefficient (P) value was needed, it was calculated by the following equation:(4)$$\ln\frac{{\left[In\right]}_{f,t}}{{\left[In\right]}_{f,0}}=-\frac{A}{V}Pt$$
where A was the membrane area, V was the volume of the feed phase and t represented the elapsed time.

## 3. Results and Discussion

The transport of indium(III) across the membrane containing the ionic liquid phase can be described by applying Fick’s first diffusion law to the diffusion layer at the feed-phase side, to the membrane phase, and to the receiving phase; however, this last contribution is often negligible compared with that at the feed phase side, as the distribution coefficient of indium(III) between the membrane and the receiving phases used to be much lower than the value between the feed and the membrane phases. Figure 1 shows the concentration profiles of indium(III) and the pseudo-protic ionic liquids dissolved in Solvesso 100 across the supported liquid membrane. Accordingly, in the transport of indium(III), the driving force was the difference in acidity between the feed and receiving phases.

### 3.1. The System In(III)-TOAH^+^Cl^−^

#### 3.1.1. Influence of Stirring Speeds in the Feed and Receiving Phases

A series of experiments were carried out to establish adequate hydrodynamic conditions both in the feed and the receiving phases. The transport of indium(III) across the supported liquid membrane was dominated by diffusional resistances which were of two types: (i) the resistance associated with the feed phase boundary layer, and (ii) the resistance associated with the membrane support. It was not rare that the magnitude of the first competed with the value of the support resistance [22]. Figure 2 shows the variation of ln([In]_f,t_/[In]_f,0_) versus time for selected stirring speeds applied in the feed phase, whereas Table 1 showed the percentage of indium transport for each stirring speed.

Maximum metal transport was obtained at 750 min^−1^ and then decreased. At 750 min^−1^, the thickness of the feed phase diffusion layer and the aqueous resistance to mass transfer were minimized, and the diffusion contribution of the aqueous species to the mass transfer phenomena was considered constant [23]. At stirring speeds above 750 min^−1^ the percentage of indium transport decreased; this situation is attributable to an increment in the turbulence caused by the stirring speed, which also results in a probable displacement of the carrier phase from the support. The value of the maximum percentage of indium transport (83%) corresponded to a limiting permeability (P_lim_) value defined as [24]:(5)$$P_{\lim}=\frac{D_{aq}}{d_{f}}$$
where D_aq_ represented the value of the aqueous diffusion coefficient with an estimated value of 10^−5^ cm^2^/s, and d_f_ was considered the minimum thickness of the feed aqueous layer. According to Equation (3), P_lim_ was calculated as 3.0 × 10^−3^ cm/s, and thus, d_f_ value was estimated as 3.3 × 10^−3^ cm. Experimental results also indicated that the percentage of indium recovered in the receiving phase reached 90% (at 3 h), indicating the usefulness of the 0.1 M HCl solution as the receiving phase.

The variation of the stirring speed in the receiving phase (500 to 750 min^−1^) had a negligible influence on the variation of the percentage of metal transport. In the case of the receiving phase, and if the stirrer in the half-cell was very close to the membrane support, the thickness of the boundary layer was considered to be minimized and the resistance in this side can be neglected [25]. Thus, stirring speeds of 750 and 500 min^−1^ were used in the feed and the receiving phases, respectively.

#### 3.1.2. Influence of HCl Concentration in the Feed Phase

Experiments were performed using organic phases of the ionic liquid dissolved in Solvesso 100 and feed phases of 0.01 g/L In(III) which also contained different HCl concentrations; Table 2 shows the results obtained. It can be seen that the presence of HCl in the feed phase tended to increase the percentage of indium transport up to 1–2 M HCl concentrations in this phase, and then decreased for higher HCl concentrations. In the first instance, this decrease can be attributed to the increase of the ionic strength in the aqueous medium.

#### 3.1.3. Influence of Carrier Concentration in the Membrane Phase

The variation in the percentage of indium transport at various carrier concentrations is shown in Figure 3. The experiments were carried out with organic phases of 2.5–40% *v*/*v* ionic liquid in Solvesso 100, and feed phases of 0.01 g/L In(III) and 1 M HCl.

The results obtained showed that increasing the carrier concentration from 2.5% *v*/*v* to 10% *v*/*v*, in the membrane phase increased the percentage of indium transport; however, from 10% *v*/*v*, this percentage decreased. This is probably attributable to the increase in viscosity of the organic phase, and as a consequence, an increase in the membrane resistance to indium transport [25,26,27].

Figure 4 shows, in the particular case using 10% *v*/*v* TOAH^+^Cl^−^ as carrier phase, the distribution of indium between the feed, membrane and receiving phases at various elapsed times. It can be seen that indium is basically not retained in the membrane phase, being in almost continuous indium flux from the feed to the receiving phase.

#### 3.1.4. Influence of the Initial Indium Concentration in the Feed Phase

Figure 5 shows the variation in the percentage of indium transport versus the initial concentration of indium, ranging from 0.01 to 0.1 g/L in the feed phase side. It can be observed that within the present experimental conditions, the percentage of metal transport decreased with the increase in the initial indium concentration in the feed phase. This behaviour is attributable to the grown population effect due to the increase in the metal concentration in the solution [28]. Additionally, the increase in the indium concentration in the feed phase resulted in the saturation of the membrane and thus, to a lower effective membrane area and to the retention of the metal-ionic liquid species on the entry side of the feed–membrane sites. Together, they caused a decrease in indium transport.

### 3.2. The System In(III)-TODAH^+^Cl^−^

Previous liquid–liquid extraction experiments demonstrated that, with this ionic liquid, maximum indium(III) extraction was obtained at 6–8 M HCl concentrations in the aqueous solution. Thus, indium transport experiments were carried out using 7 M HCl concentration in the feed solution.

#### 3.2.1. Influence of Stirring Speed in the Feed and Receiving Phases

Firstly, the stirring speed in the feed phase was investigated, while maintaining the stirring speed of the receiving phase, which was constant at 500 min^−1^. Results from this series of experiments are shown in Table 3.

It can be seen that the percentage of indium(III) transport increased up to 800–1000 min^−1^, indicating that at these stirring speeds, the thickness of the feed boundary layer reached a minimum and metal transport maximized. At higher stirring speeds, i.e., 1200 min^−1^, the percentage of transport decreased; this decrease is attributable to the same effect as that described in Section 3.1.1.

Taking into consideration Equation (5), and knowing that in the present case P_lim_ is calculated as 2.3 × 10^−3^ cm/s, the minimum thickness of the feed layer was calculated as 4.3 × 10^−3^ cm for the system In(III)-TODAH^+^Cl^−^.

Using the same experimental conditions as Table 3, and a stirring speed of 850 min^−1^ in the feed phase, while the stirring speed applied on the receiving phase was varied in the 500–750 min^−1^ range, the results showed that this variation had a minor effect on the percentage of indium transport across the supported liquid membrane. Thus, stirring speeds of 850 min^−1^ and 500 min^−1^ were used throughout the experimentation in the feed and receiving phases, respectively.

#### 3.2.2. Influence of Carrier Concentration on Indium Transport

Figure 6 represents the results obtained in this investigation about the effect of TODAH^+^Cl^−^ concentration on indium transport. The feed phase contained 0.01 g/L In(III) and 7 M HCl, whereas the corresponding organic phases were of 2.5–20% *v*/*v* ionic liquid in Solvesso 100.

From these results, it was shown that the percentage of indium transport increased with the increase in the carrier concentration, up to 10% *v*/*v*, in the organic phase immobilized on the GVHP4700 support. At carrier concentrations higher than 10% *v*/*v*, a slight decrease in metal transport was observed; this is probably due to an increase in the viscosity of the organic phase, that, as mentioned before, resulted in an increase of membrane resistance to transport.

Figure 7 shows the variation in indium distribution between the feed, membrane and receiving phases at various elapsed times. In this case, the organic phase contained 10% *v*/*v* TODAH^+^Cl^−^ in Solvesso 100, and was immobilized on a GVHP support. Again, it can be observed that there is an instantaneous indium flux from the feed to the receiving solution, demonstrating the effectiveness of the 0.1 M HCl solution in stripping indium from the metal-loaded organic phase.

#### 3.2.3. Effect of Initial Indium Concentration in the Feed Phase on Metal Transport

The influence of the initial indium concentration on the percentage of indium transport by TODAH^+^Cl^−^ was investigated. This study was performed using feed phases that contained various indium concentrations in 7 M HCl and organic phases of 10% *v*/*v* carrier in Solvesso 100. Figure 8 represents the variation of ln([In]_f,t_/[In]_f,0_) versus time for the various initial indium concentrations in the feed phase.

This figure shows how the metal transport was favoured (increased slope in the ln([In]_f,t_/[In]_f,0_) versus time plot) as the initial indium concentration in the feed phase decreased.

In terms of percentage of transport, Table 4 resumes the results. It was shown that the maximum percentage of metal transport was obtained with the feed phase containing the lowest initial indium concentration; this percentage continuously decreased as the initial metal concentration in this feed phase increased. These results can be explained by the same effects as those described in Section 3.1.4. This Table also shows that the use of 0.1 M HCl solution as receiving phase was convenient, since 80–90% of indium was recovered in the various phases.

### 3.3. Modeling of Metal Transport: Estimation of the Diffusional Parameters

#### 3.3.1. The System In(III)-TOAH^+^Cl^−^

It was described [29] that at 1 M HCl, the predominant indium(III) species in aqueous solution was InCl_3_; thus, the extraction (transport) of this metal can be expressed by the reaction
(6)$$TOAH^{+}Cl_{org}^{-}+InCl_{3_{aq}}\Leftrightarrow TOAH^{+}InCl_{4_{org}}^{-}$$
and is representative of an ion–pair extraction (transport) mechanism. The extraction constant relative to the above equilibrium can be expressed as
(7)$$K=\frac{{\left[TOAH^{+}InCl_{4}^{-}\right]}_{org}}{{\left[TOAH^{+}Cl^{-}\right]}_{org}{\left[InCl_{3}\right]}_{aq}}$$
where the subscripts org and aq are representatives of the organic phase (membrane phase) and aqueous phase (feed phase), respectively.

If the same assumptions as those described in the literature [30] were followed, an expression such as the following may be obtained:(8)$$P=\frac{K{\left[TOAH^{+}Cl^{-}\right]}_{m}}{\Delta_{m}+\Delta_{f}\left(K{\left[TOAH^{+}Cl^{-}\right]}_{m}\right)}$$
this expression combines both the equilibrium and diffusion parameters involved in the transport of In(III) from 1 M HCl solution across a membrane support containing this ionic liquid dissolved in Solvesso 100. In this Equation (8), m represented the membrane (organic phase) and f the feed (aqueous phase).

Rearranging the above expression, resulted in the following relationship:(9)$$\frac{1}{P}=\Delta_{f}+\Delta_{m}\frac{1}{K{\left[TOAH^{+}Cl^{-}\right]}_{m}}=\Delta_{f}+\Delta_{m}\frac{1}{Z}$$
where Δ_f_ and Δ_m_ were the mass transfer resistances due to the feed and the membrane phase, respectively. At various concentrations of the ionic liquid, a plot of 1/P versus 1/Z may result in a straight line with ordinate to calculate Δ_f_ and slope of Δ_m_. These values were of 177 s/cm and 1.1 s/cm for Δ_f_ and Δ_m_, respectively.

The membrane diffusion coefficient (D_m_) can be calculated according to the following expression:(10)$$D_{m}=\frac{d_{m}}{\Delta_{m}}$$
and was estimated as 1.1 × 10^−2^ cm^2^/s, considering Δ_m_= 1.1 s/cm and the thickness of the membrane support is d_m_, as 12.5 × 10^−3^ cm.

The diffusion coefficient of the indium(III)-ionic liquid species in the bulk organic phase is estimated using the following relationship [31]:(11)$$D_{b,m}=D_{m}\frac{\tau^{2}}{\varepsilon }$$
where τ is the membrane tortuosity (1.67) and ε is the support porosity (75%); thus, D_b,m_ is estimated as 4.1 × 10^−2^ cm^2^/s.

The diffusion coefficient in the bulk organic phase presented a greater value than the diffusion coefficient. This is attributable to the diffusional resistance caused by the support thickness separating the feed and receiving solutions.

Considering that the ionic liquid concentration in the membrane support is constant, the apparent diffusion coefficient for indium (III) can be calculated as:(12)$$D_{m}^{a}=\frac{Jd_{m}}{\left[R_{3}NH^{+}Cl^{-}\right]}$$
using a 10% *v*/*v* (0.23 M) ionic liquid concentration and being d_m_ 12.5 × 10^−3^ cm, this apparent diffusion coefficient has a value of 1.4 × 10^-−8^ cm^2^/s.

Finally, the mass transfer coefficient in the feed phase (Δ_f_^−1^) can be calculated as 5.6·10^−3^ cm/s.

#### 3.3.2. The System In(III)-TODAH^+^Cl^−^

In a 7 M HCL medium, the predominant indium(III) species was InCl_4_^−^ [29]. Thus, the transport of the metal can be represented by the next reaction:(13)$$TODAH^{+}Cl_{m}^{-}+InCl_{4_{f}}^{-}\Leftrightarrow TODAH^{+}InCl_{4_{m}}^{-}+Cl_{f}^{-}$$

In this case, it is representative of an anion exchange mechanism. The corresponding extraction constant is as follows:(14)$$K=\frac{{\left[TODAH^{+}InCl_{4}^{-}\right]}_{m}{\left[Cl^{-}\right]}_{f}}{{\left[TODAH^{+}Cl^{-}\right]}_{m}{\left[InCl_{4}^{-}\right]}_{f}}$$and is similar to that described in Section 3.3.1. An expression which combines both diffusional and equilibrium parameters involved in indium(III) transport can be written as:(15)$$P=\frac{K{\left[TODAH^{+}Cl^{-}\right]}_{m}{\left[Cl^{-}\right]}_{f}^{-1}}{\Delta_{m}+\Delta_{f}\left(K{\left[TODAH^{+}Cl^{-}\right]}_{m}{\left[Cl^{-}\right]}_{f}^{-1}\right)}$$
and:(16)$$\frac{1}{P}=\Delta_{f}+\Delta_{m}\frac{1}{K{\left[TODAH^{+}Cl^{-}\right]}_{m}{\left[Cl^{-}\right]}_{m}^{-1}}=\Delta_{f}+\Delta_{m}\frac{1}{Z}$$
being that Δ_f_ and Δ_m_ are calculated from the corresponding plot of 1/P and 1/Z (Figure 9).

In the present system, the values were 420 s/cm and 10200 s/cm for the resistances due to the feed phase (Δ_f_) and the membrane (Δ_m_), respectively.

Following the same series of equations than in Section 3.3.1., the values of the various diffusional parameters involved in the transport of indium(III) by TODAH^+^Cl^−^ pseudo-potric ionic liquid are resumed in Table 5.

Similar to that described in Section 3.3.1., the value of the diffusion coefficient in the bulk membrane phase was greater than the value of the membrane diffusion coefficient.

## 4. Conclusions

From the experimental data obtained from this work, it is shown that the pseudo-protic ionic liquids TOAH^+^Cl^−^ and TODAH^+^Cl^−^ can be used as carriers for indium(III) membrane transport from hydrochloric acid solutions.

For both carriers, the transport of indium is influenced by a number of variables, such as the stirring speed applied on the feed phase, and metal and carrier concentrations. In the case of TOAH^+^Cl^−^, the variation of the HCl concentration in the feed phase also influenced the transport of this strategic metal. Additionally, for both carriers, at carrier concentrations of 10% *v*/*v* in Solvesso 100, a maximum in indium(III) transport was obtained; under this condition, the transport process is controlled by diffusion in the feed phase boundary layer. At carriers concentrations lower than 10% *v*/*v*, membrane diffusion controlled the overall indium(III) transport.

Though with both ionic liquids, the driving force for indium transport is the difference in the acidities between the feed and receiving phases, based on the HCl concentration in the feed phase, the facilitated transport mechanism for indium differs from one ionic liquid to the other. At 1 M HCl concentration in the feed phase, metal is transported by an ion pair mechanism and the formation of TOAH^+^InCl_4_^−^ complex in the membrane phase. At 7 M HCl, and using TODAH^+^Cl^−^ as carrier, the complex formed in the membrane phase has the same stoichiometry, but the transport mechanism corresponded to an anionic exchange between InCl_4_^−^ of the feed phase and the chloride ions of the carrier. Thus, these chloride ions are released to the feed phase.

Experimental data indicated that 0.1 M HCl solutions can be used as effective receiving phases for both transport systems.

Various diffusional parameters are calculated for both In(III)-TOAH^+^Cl^−^ and In(III)-TODAH^+^Cl^−^ systems.

## Figures and Tables

**Figure 1 membranes-13-00019-f001:**
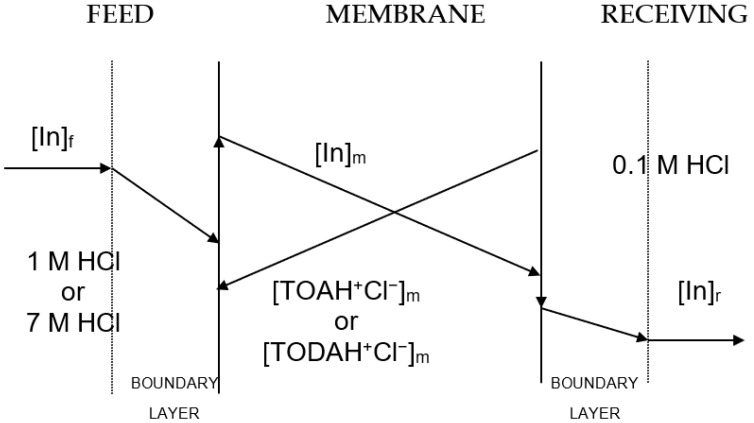
Concentration profiles of indium and PPILs species across the supported liquid membrane.

**Figure 2 membranes-13-00019-f002:**
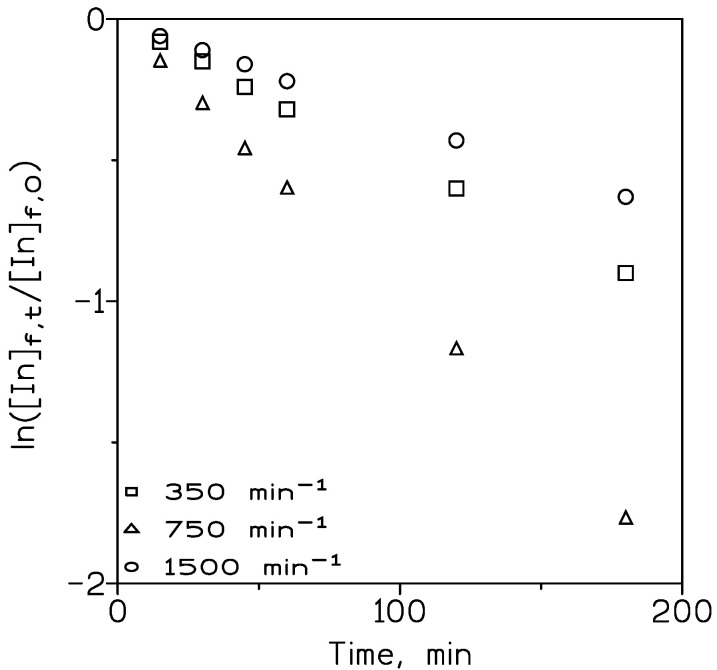
Plot of ln([In]_f,t_/[In]_f,0_) versus time at various stirring speeds. Feed phase: 0.01 g/L In(III) in 1 M HCl. Organic phase: 10% *v*/*v* TOAH^+^Cl^−^ in Solvesso 100 immobilized in GVHP support. Receiving phase: 0.1 M HCl.

**Figure 3 membranes-13-00019-f003:**
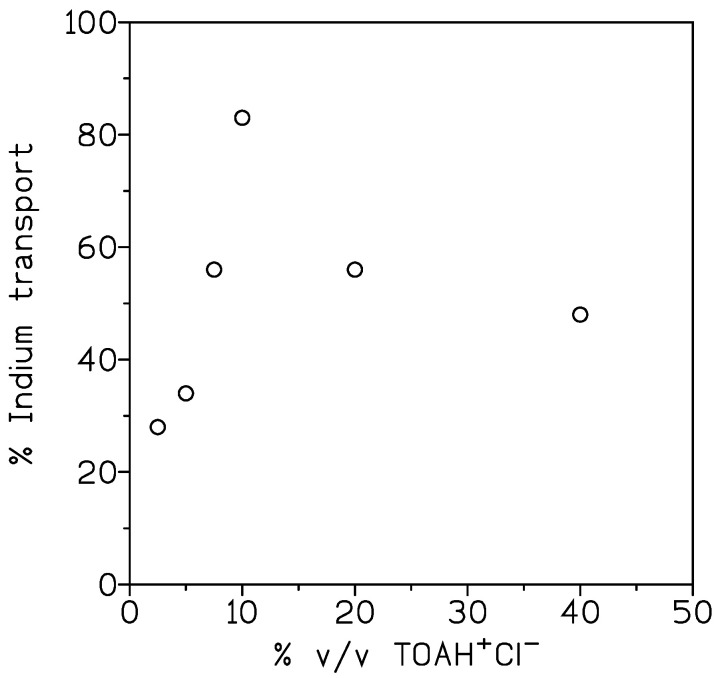
Influence of TOAH^+^Cl^−^ concentration on the transport of indium. Membrane support: GVHP4700. Receiving phase: 0.1 M HCl. Temperature: 20 °C. Time: 3 h.

**Figure 4 membranes-13-00019-f004:**
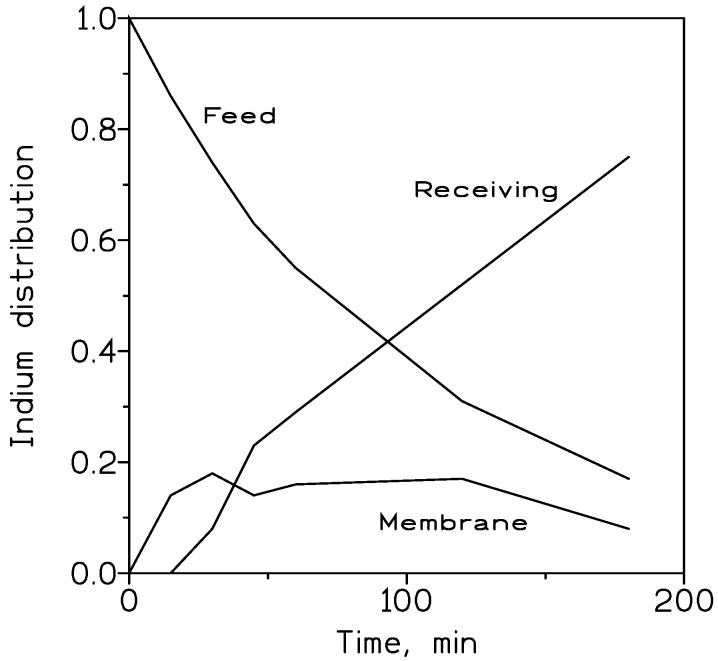
Indium distribution profile between the feed, membrane and receiving phases. Feed phase: 0.01 g/L In(III) in 1 M HCl. Organic phase: 10% *v*/*v* TOAH^+^Cl^−^ in Solvesso 100 immobilized in GVHP support. Receiving phase: 0.1 M HCl. Temperature: 20 °C.

**Figure 5 membranes-13-00019-f005:**
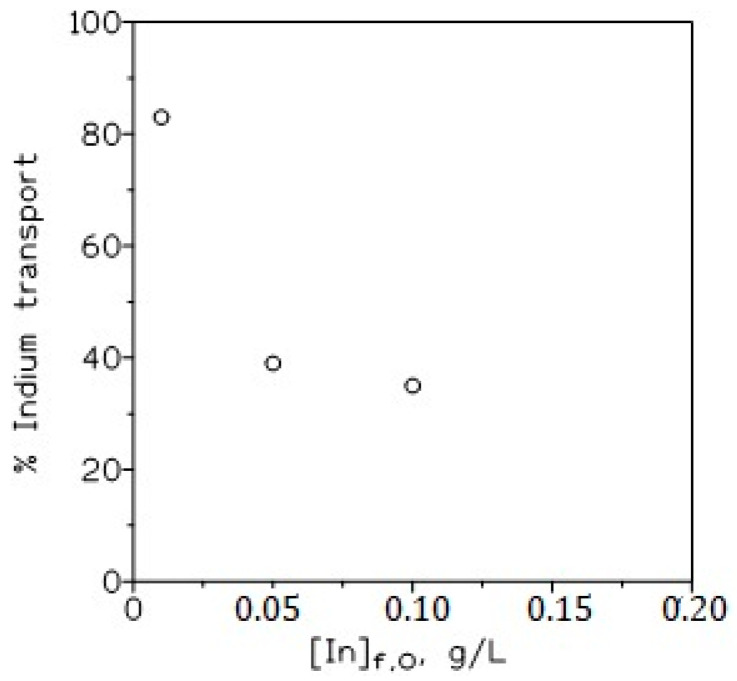
Influence of initial indium concentration on metal transport. Feed phase: indium and 1 M HCl. Membrane phase: 10% *v*/*v* TOAH^+^Cl^−^ in Solvesso 100 supported on GHVP4700. Receiving phase; 0.1 M HCl. Temperature: 20 °C. Time: 3 h.

**Figure 6 membranes-13-00019-f006:**
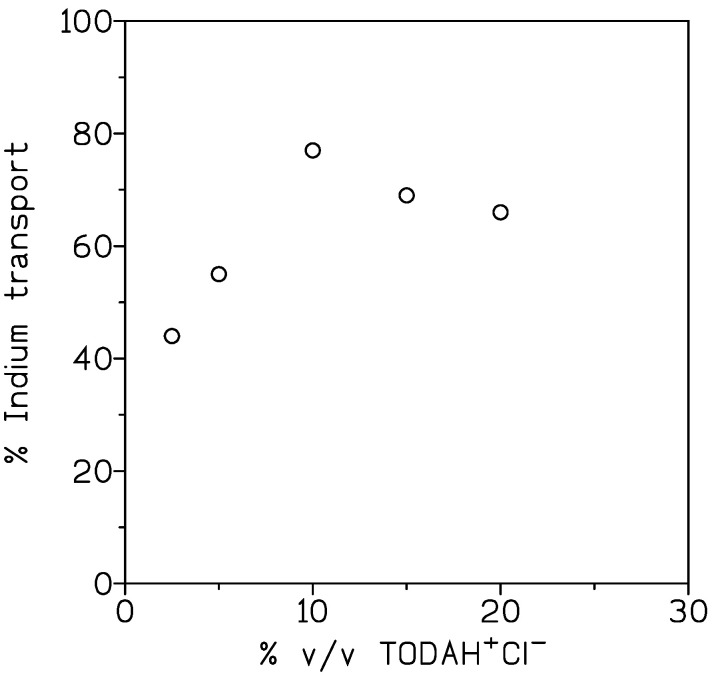
Influence of TODAH^+^Cl^−^ concentration on the transport of indium across the supported liquid membrane. Membrane support: GVHP4700. Receiving phase: 0.1 M HCl. Temperature: 20 °C. Time. 3 h.

**Figure 7 membranes-13-00019-f007:**
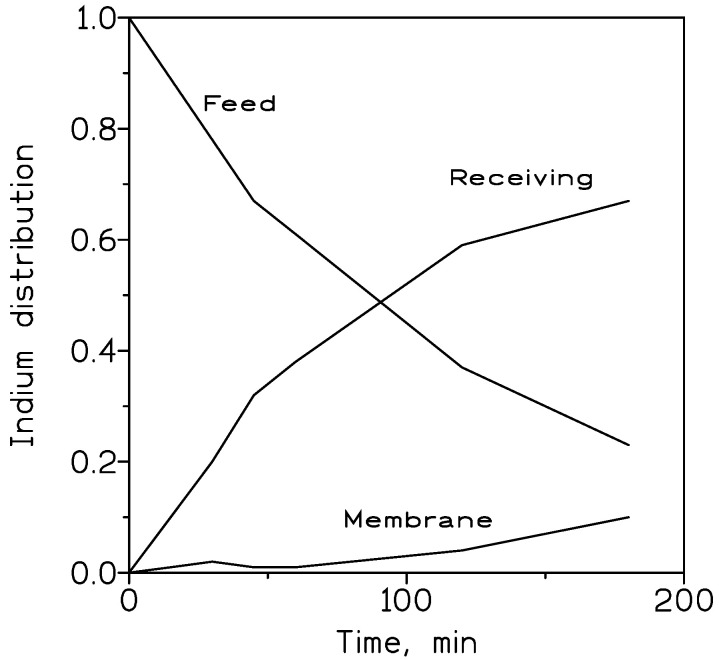
Indium distribution profile between feed, membrane and receiving phases. Feed phase: 0.01 g/L In(III) in 7 M HCl. Organic phase: 10% *v*/*v* TODAH^+^Cl^−^ in Solvesso 100 immobilized on GVHP support. Receiving phase: 0.1 M HCl. Temperature: 20 °C.

**Figure 8 membranes-13-00019-f008:**
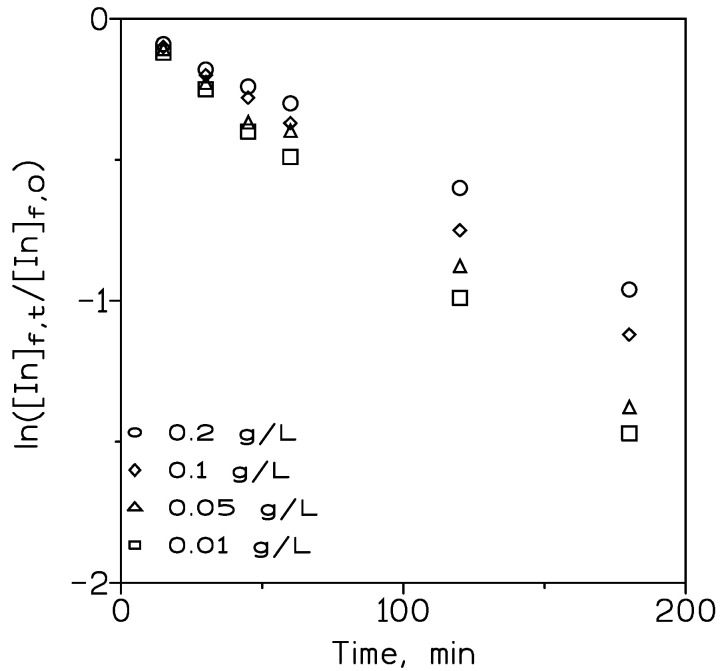
Variation of ln([In]_f,t_/[In]_f,0_) versus time for different initial indium concentrations in the feed phase. Feed phase: Indium in 7 M HCl. Organic phase: 10% *v*/*v* TODAH^+^Cl^−^ in Solvesso 100 immobilized on GVHP support. Receiving phase: 0.1 M HCl. Temperature: 20 °C.

**Figure 9 membranes-13-00019-f009:**
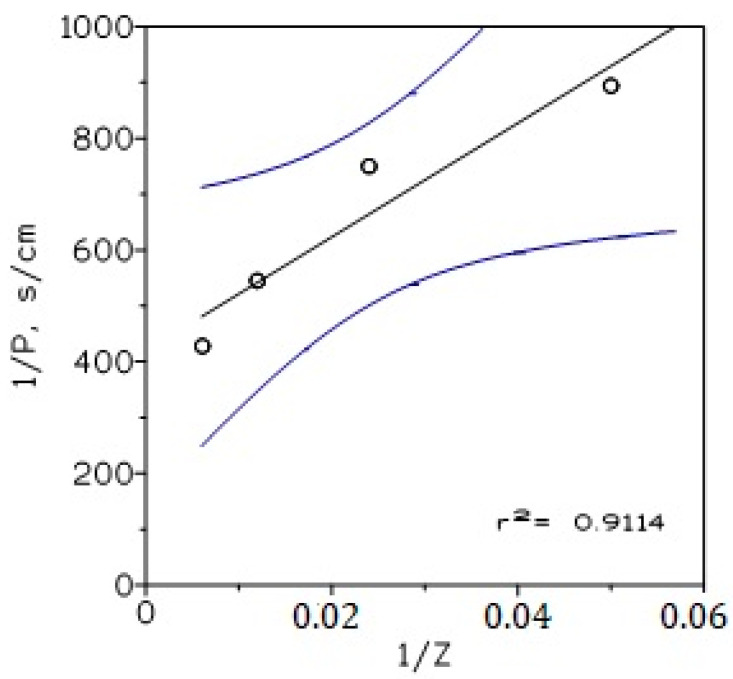
Plot of 1/P versus 1/Z (Equation (16)). Feed phase: 0.01 g/L In(III) in 7 M HCl. Organic phase: 1.25–10% *v*/*v* TODAH^+^Cl^−^ in Solvesso 100 immobilized on GVHP support. Receiving phase: 0.1 M HCl. Temperature: 20 °C. Dotted line showed 95% confidence interval of the regression line.

**Table 1 membranes-13-00019-t001:** Influence of the stirring speed in the feed phase on indium transport.

Stirring Speed, min^−1^	Percent Transport
375	64
500	73
750	83
1000	63
1500	47

Experimental conditions as in Figure 2. Receiving phase stirring speed: 500 min^−1^. Temperature: 20 °C. Time: 3 h.

**Table 2 membranes-13-00019-t002:** Influence of the HCl concentration on indium transport.

HCl, M	Percent Transport
0.5	25
0.75	50
1	83
2	83
4	68
6	60

Membrane phase: 10% *v*/*v* TOAH^+^Cl^−^ in Solvesso 100 supported on GVHP4700. Receiving phase: 0.1 M HCl. Temperature: 20 °C. Time 3 h.

**Table 3 membranes-13-00019-t003:** Influence of the stirring speed on indium transport.

Stirring Speed, min^−1^	Percentage Transport
600	55
800	77
1000	77
1200	66
1500	60

Feed phase: 0.01 g/L In(III) and 7 M HCl. Membrane phase: 10% *v*/*v* TODAH^+^Cl^−^ in Solvesso 100 supported on GHVP4700. Receiving phase: 0.1 M HCl. Temperature: 20 °C. Time: 3 h.

**Table 4 membranes-13-00019-t004:** Influence of initial metal concentration on indium transport.

Initial Indium(III) Concentration, g/L	Percent Transport Feed Phase to Membrane Phase	Percent Transport Membrane Phase to Receiving Phase
0.01	77	87
0.05	74	88
0.1	67	90
0.2	62	82

Membrane support: GVHP4700. Receiving phase: 0.1 M HCl. Temperature: 20 °C. Time: 3 h

**Table 5 membranes-13-00019-t005:** Diffusional parameters in the In(III)-TODAH^+^Cl^−^ system.

Parameter	Value
Membrane diffusion coefficient (D_m_)	1.2 × 10^−6^ cm^2^/s
Diffusion coefficient in the bulk membrane (D_b,m_)	4.5 × 10^−6^ cm^2^/s
Apparent diffusion coefficient in the membrane (D_m_^a^)	1.2 × 10^−8^ cm^2^/s
Mass transfer coefficient in the feed phase (Δ_f_^−1^)	2.3 × 10^−3^ cm/s

Values at the maximum indium transport at 7 M HCl in the feed phase.

## Data Availability

Not applicable.

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
