# Peer review of "The Pseudo-Protic Ionic Liquids TOAH+Cl and TODAH+Cl as Carriers for Facilitated Transport of In(III) from HCl Solutions"

_membranes, 2022, doi:10.3390/membranes13010019_

Round 1

Reviewer 1 Report

Membranes

Manuscript ID: membranes-2069771

Authors: Francisco José Alguacil , Félix Antonio López *

Title: The pseudo-protic ionic liquids TOAH+Cl- and TODAH+Cl- as carriers for facilitated transport of In(III) from HCl solutions

The present work describes the behavior of the pseudo-protic ionic liquids TOAH+Cl- and TODAH+Cl- in the facilitated transport of indium (III) from HCl solutions by the supported liquid membrane (SLM).

Authors show that the pseudo-protic liquids can be used as carriers for In (III) ions membrane transport from hydrochloric acid solutions. Transport of the In (III) ion is investigated as a function of hydrodynamic conditions, HCl and In (III) ion concentration in the feed phase, and carrier concentration in the membrane phase. Moreover, various diffusional parameters are calculated for both carriers.

This work is evidently interesting. The respective experiments were done according to present-date standards. The results are correctly described so that they seem to be of interest to the readership of Membranes. Therefore, the manuscript should be published in this Journal.

However, before publication, some important observations require further discussion. Moreover, the scope of research should be extended.

First of all, it is not explained how the authors found that the process is controlled by diffusion and not by chemical reaction. So the question is, how does the constant rate be dependent on temperature?

Secondly, the authors did not mention the most critical issue regarding SLM, namely, the stability of the membrane.

In the chapter on the influence of the mixing speed on the efficiency of the process (3.1.1), the authors indicate that at higher rates, there may be a "loss" of the membrane phase from the support. Many authors take this as the main cause of SLM instability. Will the most favorable extraction parameters selected by the authors allow for the maintenance of the selectivity and effectiveness of the membrane in repeated cycles? This has not been tested, although it is a crucial issue. Moreover, the authors point in the introduction section to the stability issues of the carriers used (pages 2 and 3, lines 46-47 and 103-105). That should be the primary goal of the work.

I understand that the described problem of In( III) recovery is very current. This has been very well supported by a reliable review of the latest literature. However, repeating the tests of various carriers and demonstrating their satisfactory efficiency and selectivity (https://doi.org/10.3390/molecules25225238) will not enable the practical implementation of liquid membranes.

The comments included in the attached file are recommended that the original content of this paper will be valuable for publication.

Author Response

The present work describes the behavior of the pseudo-protic ionic liquids TOAH+Cl- and TODAH+Cl- in the facilitated transport of indium (III) from HCl solutions by the supported liquid membrane (SLM).

Authors show that the pseudo-protic liquids can be used as carriers for In (III) ions membrane transport from hydrochloric acid solutions. Transport of the In (III) ion is investigated as a function of hydrodynamic conditions, HCl and In (III) ion concentration in the feed phase, and carrier concentration in the membrane phase. Moreover, various diffusional parameters are calculated for both carriers.

This work is evidently interesting. The respective experiments were done according to present-date standards. The results are correctly described so that they seem to be of interest to the readership of Membranes. Therefore, the manuscript should be published in this Journal.

However, before publication, some important observations require further discussion. Moreover, the scope of research should be extended.

First of all, it is not explained how the authors found that the process is controlled by diffusion and not by chemical reaction. So the question is, how does the constant rate be dependent on temperature?

Accordingly with the theory when a transport system reached a maximum in permeability, the thickness of the aqueous boundary layer is a minimum, and the transport is dominated by diffusion, in other conditions, i.e. lower carrier concentrations, membrane diffusion is the controlling step.

About your question in relation with temperature and constant rate, this relation can be only demonstrated under experimentation. At a first instance, an increase in the temperature decreased the viscosity and the system may permeate better.     

Secondly, the authors did not mention the most critical issue regarding SLM, namely, the stability of the membrane.

Much has been theorized about SLM stability. Like in other separations technologies, i.e. liquid-liquid extraction, if the operation is properly done, the stability of the system is good. In any case, SLM in flat-sheet operation is not the best system to investigate about the membrane stability. This has only sense when the system is scale-up to hollow fiber devices, and working over an extended time period, again like in liquid-liquid extraction: batch against mixer-settlers operation.

In the chapter on the influence of the mixing speed on the efficiency of the process (3.1.1), the authors indicate that at higher rates, there may be a "loss" of the membrane phase from the support. Many authors take this as the main cause of SLM instability. Will the most favorable extraction parameters selected by the authors allow for the maintenance of the selectivity and effectiveness of the membrane in repeated cycles? This has not been tested, although it is a crucial issue. Moreover, the authors point in the introduction section to the stability issues of the carriers used (pages 2 and 3, lines 46-47 and 103-105). That should be the primary goal of the work.

As we wrote above, experimentation in a cell is not (in our opinion) the best method to check the membrane stability. And allow us to disappoint with you, the primary goal of this work, all the works, is to investigate the parameters which affect the respective system and if the system is useful for i.e. indium transport, surprisingly one can find that problems encountered in batch operation do not appear in continuous operation, and the reversal, problems arising in continuous operation had not been  predicted in batch operation. The above though also served for all the separation technologies that you imagine (liquid-liquid extraction, adsorption, filtration, etc.     

I understand that the described problem of In( III) recovery is very current. This has been very well supported by a reliable review of the latest literature. However, repeating the tests of various carriers and demonstrating their satisfactory efficiency and selectivity (https://doi.org/10.3390/molecules25225238) will not enable the practical implementation of liquid membranes.

We agreed with you

Please find below our responses to the questions raised by you on the pdf version of the manuscript. Mention to lines refereed to the original manuscript version.

Abstract. Lines 10-14.

We rewrite the abstract section

Introduction. Line 36.

In our opinion these separation technologies are well known, this further information, and or inclusion of application references seemed a little bit redundant and offer nothing of interest to the manuscript

Line 38.

We delete the sentence

Line 40.

We rewrite the sentence

Line 48.

Reference included

Line 51.

Please note that in the Introduction section of our works, all the references related with the given theme (in the present case separation technologies applied to indium recovery) if possible they are not older than two-three years, thus, please accept that we do not include that suggested by yours (dated back 2017).

Line 100.

Amended

Line 103.

A comment has been added

Line 107.

There was not any special expectation. Simply, these were two amines which we had in our laboratory. We think that the information requested by you is not of scientific value

Line 125.

These conclusions are based in our extended experience (over 40 years) in the use of liquid-liquid extraction operation. Thus, these are conclusions by our own, no references here

Lines 128 to 130.

Please note that the information that you request about the membrane support is given in line 130 of the original manuscript. In our opinion, the information about In(III) concentrations is irrelevant in the context of the manuscript

Line 139.

Previous tests mean that we perform several previous experiments (not published) to fix the immersion time. No references here

Line 161.

We do not agree with you, we think that here is the proper place to put this paragraph 

Line 173.

This is based in our experience and what the many experimental results demonstrated. Please note that reference 19 (old manuscript), 21 in the revised version, is in connection with the above.

Line 179.

This is an ordinary running time for our experiments

Line 181.  

Reference added

Eq.(4) in the original manuscript

Reference added

Line 201.

We think that in the Material section this information must not be included

Line 208.

We think that a reference here is not critical

Line 214.

Amended

Line 216.

We corrected the location of Table 2

Line 246.

Reference added

Line 254.

Probably not, but it needs to be experimentally demonstrated

Line 267.

Yes, however in our opinion generalization here is difficult to achieve

Line 272.

As it is previously mentioned, previous experiments mean experiments done in the experimental context of the work but not published. They are done just to fix conditions.

Conclusions

We add a mention about stripping, as well as along the manuscript   

Reviewer 2 Report

1. The Abstract should briefly state the purpose of the research, principal results and major conclusions. The authors should rewrite the abstract for these standards.

2. The English style in the manuscript should be improved with the help of native skilled English speakers for a vintage look.

3. The introduction has to be rewritten. Authors have to avoid long, complicated sentences. More references have to be added. The introduction has to show a huge number of recent research that has been done in this sector.

Line 21-29, for example.

Lines 33-37 are hard to read + incorrect spelling and punctuation, so I would advise you to avoid hefty sentences. In the case of using (i), (ii), (iii), instead use numbers or letters and put them one after one, not in the text. Before (i), (ii), (iii) have to be a colon (:).

Line 39-41, unclear sentences must be rewritten and clarified.

Line 100-105 unclear sentences must be rewritten and clarified.

4. Methods.  In this section, the figure of the device where the performed research was conducted must be shown.

5. Results and discussion
3.1.3 Influence of carrier concentration in the membrane phase
Line 213 Fig. 1 or Fig. 2?
3.2.1 . Influence of stirring speed in the feed and receiving phases
Lines 285 - 287 clarify
I would suggest comparing the Influence of the initial indium concentration in the feed phase and membrane for TOAH+Cl- and TODAH+Cl- in one section. To divide them into four different sections is unnecessary. Comparing them in one section will avoid misunderstanding for the reader.

6. The conclusion clearly describes the achieved results. It has to be in connection with the abstract. Where the Abstract should be mentioned the main results that will be performed, and then, in conclusion, these results are shown. The conclusion and Abstract should be compared and rewritten. 

Author Response

  1. The Abstract should briefly state the purpose of the research, principal results and major conclusions. The authors should rewrite the abstract for these standards.

Done

  1. The English style in the manuscript should be improved with the help of native skilled English speakers for a vintage look.

Done. The person in question declares that the English is pretty understandable

  1. The introduction has to be rewritten. Authors have to avoid long, complicated sentences. More references have to be added. The introduction has to show a huge number of recent research that has been done in this sector.

Done. However we do not agree with you respect to the references. A manuscript is not better based in the number of references, thus, a manuscript must not have a huge number of references as you said, unless the manuscript is a review. We checked references published in 2022 about the use of membranes on metal recovery, and believe it or not, the inclusion of these will  add nothing of interest to the investigation presented in our manuscript.    

Line 21-29, for example.

Lines 33-37 are hard to read + incorrect spelling and punctuation, so I would advise you to avoid hefty sentences. In the case of using (i), (ii), (iii), instead use numbers or letters and put them one after one, not in the text. Before (i), (ii), (iii) have to be a colon (:).

We do not agree with you. The numeration of i), ii) etc, is not rare in scientific literature, we use it several times.

Line 39-41, unclear sentences must be rewritten and clarified.

Done

Line 100-105 unclear sentences must be rewritten and clarified.

Done

  1. Methods. In this section, the figure of the device where the performed research was conducted must be shown.

We think that the inclusion of such figure is not necessary, since the configuration of a permeation cell is well know.. However, and in the case that you have any doubt, and for your information, we included it here.

  1. Results and discussion

3.1.3 Influence of carrier concentration in the membrane phase

Line 213 Fig. 1 or Fig. 2?

Fig, 3 is now correct

3.2.1 . Influence of stirring speed in the feed and receiving phases

Lines 285 - 287 clarify

Done

I would suggest comparing the Influence of the initial indium concentration in the feed phase and membrane for TOAH+Cl- and TODAH+Cl- in one section. To divide them into four different sections is unnecessary. Comparing them in one section will avoid misunderstanding for the reader.

We do not agree with you, and since the conditions are not the same,  in our opinion it is better to consider both systems separately.

  1. The conclusion clearly describes the achieved results. It has to be in connection with the abstract. Where the Abstract should be mentioned the main results that will be performed, and then, in conclusion, these results are shown. The conclusion and Abstract should be compared and rewritten.

Done

Reviewer 3 Report

This paper deals with indium permeation through the SLM including TOAHCl or TODAHCl. The following points should be considered.

1. Introduction:  There are many papers on the indium extraction and liquid membranes including quaternary ammonium salts. Authors should describe these studies in Introduction.

2. Experimental: Time courses of indium concentrations in feed and receiving phases and plots based on Eq. (3) should be shown. 

3. Results and Discussion

3.1 Figs. 2 and 4. There are no viscosity data. Please show the relation between viscosity of the solvent and concentration.

3.2 Please show the plots based on Eq. (15).

3.3 Parameters listed in Table 5 should be discussed in comparison with previously reported values.

3.4 Please indicate the advantages of using pseudo-ionic liquids obtained in this study over previous studies.

Author Response

This paper deals with indium permeation through the SLM including TOAHCl or TODAHCl. The following points should be considered.

  1. Introduction: There are many papers on the indium extraction and liquid membranes including quaternary ammonium salts. Authors should describe these studies in Introduction.

It is our style of writing only include recent references (i.e. one or two years old) 

  1. Experimental: Time courses of indium concentrations in feed and receiving phases and plots based on Eq. (3) should be shown.

We add an example, please note that if we include all the data that you comment, the manuscript will present a huge number of figures.

  1. Results and Discussion

3.1 Figs. 2 and 4. There are no viscosity data. Please show the relation between viscosity of the solvent and concentration.

We add references

3.2 Please show the plots based on Eq. (15).

Done

3.3 Parameters listed in Table 5 should be discussed in comparison with previously reported values.

I our opinion, this comparison is hard to be done since the experimental conditions and mechanisms are not the same.

3.4 Please indicate the advantages of using pseudo-ionic liquids obtained in this study over previous studies.

This is only a preliminary investigation, thus, at this moment this comparison is simply speculative, also because the experimental conditions, in which the experimentations had been done, were not the same. 

Round 2

Reviewer 1 Report

The author has responded well to the reviewer's questions, and the paper can be accepted in present form.

Reviewer 3 Report

The manuscript is satisfactorily revised for publication.